# Model-based spatial-temporal mapping of opisthorchiasis in endemic countries of Southeast Asia

**Ting-Ting Zhao[1], Yi-Jing Feng[1], Pham Ngoc Doanh[2], Somphou Sayasone[3], Virak Khieu[4], Choosak Nithikathkul[5], Men-Bao Qian[6,7], Yuan-Tao Hao[1,8], Ying-Si Lai[1,8]\***

[1]Department of Medical Statistics, School of Public Health, Sun Yat-sen University, Guangzhou, China; [2]Department of Parasitology, Institute of Ecology and Biological Resources, Graduate University of Science and Technology, Vietnam Academy of Sciences and Technology, Cau Giay, Hanoi, Viet Nam; [3]Lao Tropical and Public Health Institute, Ministry of Health, Vientiane, Lao People's Democratic Republic; [4]National Center for Parasitology, Entomology and Malaria Control, Ministry of Health, Phnom Penh, Cambodia; [5]Tropical and Parasitic Diseases Research Unit, Faculty of Medicine, Mahasarakham University, Mahasarakham, Thailand; [6]National Institute of Parasitic Diseases, Chinese Center for Disease Control and Prevention, Shanghai, China; [7]WHO Collaborating Centre for Tropical Diseases, Key Laboratory of Parasite and Vector Biology, Ministry of Health, Shanghai, China; [8]Sun Yat-sen Global Health Institute, Sun Yat-sen University, Guangzhou, China

**Abstract** Opisthorchiasis is an overlooked danger to Southeast Asia. High-resolution disease risk maps are critical but have not been available for Southeast Asia. Georeferenced disease data and potential influencing factor data were collected through a systematic review of literatures and open-access databases, respectively. Bayesian spatial-temporal joint models were developed to analyze both point- and area-level disease data, within a logit regression in combination of potential influencing factors and spatial-temporal random effects. The model-based risk mapping identified areas of low, moderate, and high prevalence across the study region. Even though the overall population-adjusted estimated prevalence presented a trend down, a total of 12.39 million (95% Bayesian credible intervals [BCI]: 10.10–15.06) people were estimated to be infected with *O. viverrini* in 2018 in four major endemic countries (i.e., Thailand, Laos, Cambodia, and Vietnam), highlighting the public health importance of the disease in the study region. The high-resolution risk maps provide valuable information for spatial targeting of opisthorchiasis control interventions.

**\*For correspondence:**
laiys3@mail.sysu.edu.cn

**Competing interests:** The authors declare that no competing interests exist.

## Introduction

End of the epidemics of neglected tropical diseases (NTDs) by 2030 embodied in the international set of targets for the sustainable development goals (SDGs) endorsed by the United Nations empowers the efforts made by developing countries to combat the NTD epidemics (*UN, 2015*). To date, 20 diseases have been listed as NTDs, and opisthorchiasis is under the umbrella of food-borne trematodiasis (*Ogorodova et al., 2015*). Two species of opisthorchiasis are of public health significance, that is, *Opisthorchis felineus* (*O. felineus*), endemic in eastern Europe and Russia, and *Opithorchis viverrini* (*O. viverrini*), endemic in Southeast Asian countries (*Petney et al., 2013*). The later species is of our interest in the current article.

According to WHO's conservative estimation, an overall disease burden due to opisthorchiasis was 188,346 disability-adjusting life years (DALYs) in 2010 (*Havelaar et al., 2015*). Fürst and colleagues estimated that more than 99% of the burden worldwide attribute to *O. viverrini* infection in Southeast Asia (*Fürst et al., 2012*). Five countries in Southeast Asia, Cambodia, Lao PDR, Myanmar, Thailand, and Vietnam, are endemic for opisthorchiasis, with an estimated 67.3 million people at risk (*Keiser and Utzinger, 2005*). It is well documented that chronic and repeated infection with *O. viverrini* leads to the development of fatal bile duct cancer (cholangiocarcinoma) (*International Agency for Research on Cancer, 1994*).

The life cycle of *O. viverrini* involves freshwater snails of the genus Bithynia as the first intermediate host, and freshwater cyprinoid fish as the second intermediate host. Humans and other carnivores (e.g., cats and dogs), the final hosts, become infected by consuming raw or insufficiently cooked infected fish (*Andrews et al., 2008*; *Saijuntha et al., 2014*). Behavioral, environmental, and socioeconomic factors affect the transmission of *O. viverrini* (*Grundy-Warr et al., 2012*, *Phimpraphai et al., 2017*, *Phimpraphai et al., 2018*, *Prueksapanich et al., 2018*). Raw or insufficiently cooked fish consumption is the cultural root in endemic countries, showing a strong relationship with the occurrence of the disease (*Andrews et al., 2008*; *Grundy-Warr et al., 2012*). Poorly hygienic conditions increase the risk of infection, especially in areas practicing raw-fish-eating habit (*Grundy-Warr et al., 2012*). In addition, environmental and climatic factors, such as temperature, precipitation, and landscape, affecting either snail/fish population or growth of the parasites inside the intermediate hosts, can potentially influence the risk of human infection (*Forrer et al., 2012*; *Suwannatrai et al., 2017*). Important control strategies of *O. viverrini* infection include preventive chemotherapy, health education, environmental modification, improving sanitation, as well as comprehensive approaches with combinations of the above (*Saijuntha et al., 2014*). For purposes of public health control, WHO recommends implementing preventive chemotherapy once a year or once every 2 years depending on the levels of prevalence in population, with complementary interventions such as health education and improvement of sanitation (*WHO, 2009*).

Understanding the geographical distribution of *O. viverrini* infection risk at high spatial resolution is critical to prevent and control the disease cost-effectively in priority areas. Thailand conducted national surveys for *O. viverrini* prevalence in 1981, 1991, 2001, 2009, and 2014 (*Echaubard et al., 2016*; *Suwannatrai et al., 2018*), but the results of these surveys were presented at the province level, which is less informative for precisely targeting control interventions. Suwannatrai and colleagues, based on climatic and *O. viverrini* presence data, produced climatic suitability maps for *O. viverrini* in Thailand using the MaxEnt modeling approach (*Suwannatrai et al., 2017*). The maps brought insights for identifying areas with a high probability of *O. viverrini* occurrence; however, they did not provide direct information on prevalence of *O. viverrini* in population (*Elith et al., 2011*). A risk map of *O. viverrini* infection in Champasack province of Lao PDR was presented by Forrer and colleagues (*Forrer et al., 2012*). To our knowledge, high-resolution, model-based risk estimates of *O. viverrini* infection are unavailable in the whole endemic region of Southeast Asia.

Bayesian geostatistical modeling is one of the most rigorous inferential approaches for high-resolution maps depicting the distribution of the disease risk (*Karagiannis-Voules et al., 2015*). Geostatistical modeling relates geo-referenced disease data with potential influencing factors (e.g., socioeconomic and environmental factors) and estimates the infection risk in areas without observed data (*Gelfand and Banerjee, 2017*). Common geostatistical models are usually based on point-referenced survey data (*Banerjee et al., 2014*). In practice, disease data collected from various sources often consists of point-referenced and area-aggregated data. Bayesian geostatistical joint modeling approaches provide a flexible framework for combining analysis of both kinds of data (*Moraga et al., 2017*; *Smith et al., 2008*). In this study, we aimed (1) to collect all available survey data on the prevalence of *O. viverrini* infection at point- or area-level in Southeast Asia through systematic review; and (2) to estimate the spatial-temporal distribution of the disease risk at a high spatial resolution, with the application of advanced Bayesian geostatistical joint modeling approach.

## Results

A total of 2690 references were identified through systematically reviewing peer-review literatures, and 13 additional references were gathered from other sources. According to the inclusion and exclusion criteria, 168 records were included, resulted in a total of 580 ADM1-level surveys in 174

areas, 210 ADM2-level surveys in 142 areas, 53 ADM3-level surveys in 51 areas, and 251 point-level surveys at 207 locations in five endemic countries (i.e., Cambodia, Lao PDR, Myanmar, Thailand, and Vietnam) of Southeast Asia (*Figure 1*). Around 70% and 15% of surveys were conducted in Thailand and Lao PDR, respectively. Only two relevant records were obtained from Myanmar. To avoid large estimated errors, we did not include this data in the final geostatistical analysis. All surveys were

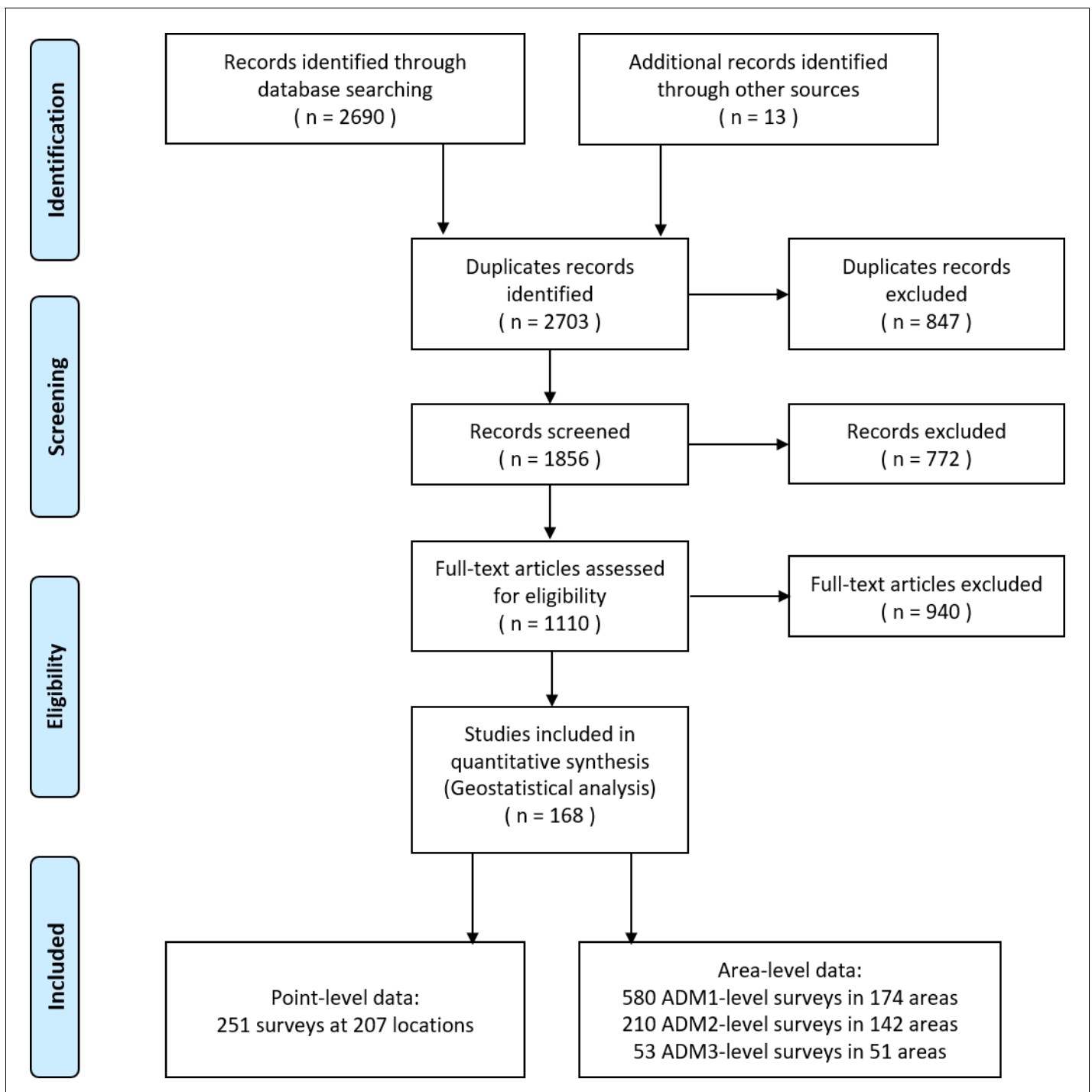

**Figure 1.** Data search and selection flow chart.

The online version of this article includes the following figure supplement(s) for figure 1:

**Figure supplement 1.** Study protocol.

conducted after 1970, with around 75% done after 1998. Most surveys (95%) are community based. Around 40% of surveys used the Kato–Katz technique for diagnosis, while another 42% did not specify diagnostic approaches. Mean prevalence calculated directly from survey data was 16.74% across the study region. A summary of survey data is listed in *Table 1*, and survey locations and observed prevalence in each period are shown in *Figure 2*. Area-level data cover all regions in Thailand and Lao PDR, and most regions in Cambodia and Vietnam, while point-referenced data are absent in most areas of Vietnam, the western part of Cambodia and southern part of Thailand. Around 70% of eligible literatures got a score equal or more than 7, indicating an overall good quality of eligible literatures in our study (*Figure 2—figure supplement 1*).

Seven variables were selected for the final model through the Bayesian variable selection process (*Table 2*). The infection risk was 2.61 (95% BCI: 2.10–3.42) times in the community as much as that in school-aged children. Surveys using FECT (formalin-ethyl acetate concentration technique) as the diagnostic method showed a lower prevalence (*OR* 0.76, 95% BCI: 0.61–0.93) compared to that using Kato–Katz method, while no significant difference was found between Kato–Katz and the other diagnostic methods. Human influence index and elevation were negatively correlated with the infection risk. Each unit increase of the HII index was associated with 0.01 (95% BCI: 0.003–0.02) decrease

**Table 1.** Overview of opisthorchiasis survey data in Southeast Asia.

| | Cambodia | Lao PDR | Myanmar | Thailand | Vietnam | Total |
|---|---|---|---|---|---|---|
| Relevant papers | 14 | 43 | 2 | 97 | 15 | 168 |
| Total surveys/locations | 91/73 | 156/99 | 6/6 | 770/335 | 71/64 | 1094/574 |
| Survey type (surveys/locations) | | | | | | |
| School | 33/31 | 4/4 | 0/0 | 13/13 | 0/0 | 50/48 |
| Community | 58/46 | 152/94 | 6/6 | 757/325 | 71/64 | 1044/535 |
| Location type (surveys/locations) | | | | | | |
| Point-level | 55/43 | 63/51 | 3/3 | 125/105 | 5/5 | 251/207 |
| ADM3-level | 0/0 | 0/0 | 0/0 | 53/51 | 0/0 | 53/51 |
| ADM2-level | 14/11 | 35/27 | 0/0 | 159/102 | 2/2 | 210/142 |
| ADM1-level | 22/19 | 58/18 | 3/3 | 433/77 | 64/57 | 580/174 |
| Period | 1998–2016 | 1989–2016 | 2015–2016 | 1978–2018 | 1991–2015 | 1978–2018 |
| Year of survey (surveys/locations) | | | | | | |
| 1978–1982 | 0/0 | 0/0 | 0/0 | 123/115 | 0/0 | 123/115 |
| 1983–1987 | 0/0 | 0/0 | 0/0 | 7/6 | 0/0 | 7/6 |
| 1988–1992 | 0/0 | 2/2 | 0/0 | 97/89 | 1/1 | 100/92 |
| 1993–1997 | 0/0 | 9/5 | 0/0 | 18/18 | 6/2 | 33/25 |
| 1998–2002 | 25/22 | 28/22 | 0/0 | 103/103 | 2/2 | 158/149 |
| 2003–2007 | 3/2 | 26/24 | 0/0 | 15/15 | 1/1 | 45/42 |
| 2008–2012 | 62/48 | 75/54 | 0/0 | 166/153 | 9/8 | 312/263 |
| 2013–2018 | 1/1 | 16/16 | 6/6 | 241/201 | 52/52 | 316/276 |
| Diagnostic methods (surveys/locations) | | | | | | |
| Kato–Katz | 86/70 | 128/83 | 3/3 | 212/166 | 7/7 | 436/329 |
| FECT | 2/2 | 8/7 | 3/3 | 109/99 | 0/0 | 122/111 |
| Stoll's | 0/0 | 0/0 | 0/0 | 38/28 | 0/0 | 38/28 |
| PCR | 0/0 | 5/4 | 0/0 | 1/1 | 0/0 | 6/5 |
| Combined | 3/3 | 14/13 | 0/0 | 14/12 | 0/0 | 31/28 |
| Others | 0/0 | 1/1 | 0/0 | 6/6 | 0/0 | 7/7 |
| NS[*] | 0/0 | 5/5 | 0/0 | 391/111 | 64/57 | 460/173 |
| Mean prevalence | 10.56% | 39.50% | 4.93% | 14.25% | 2.65% | 16.74% |

*NS: not stated or missing.

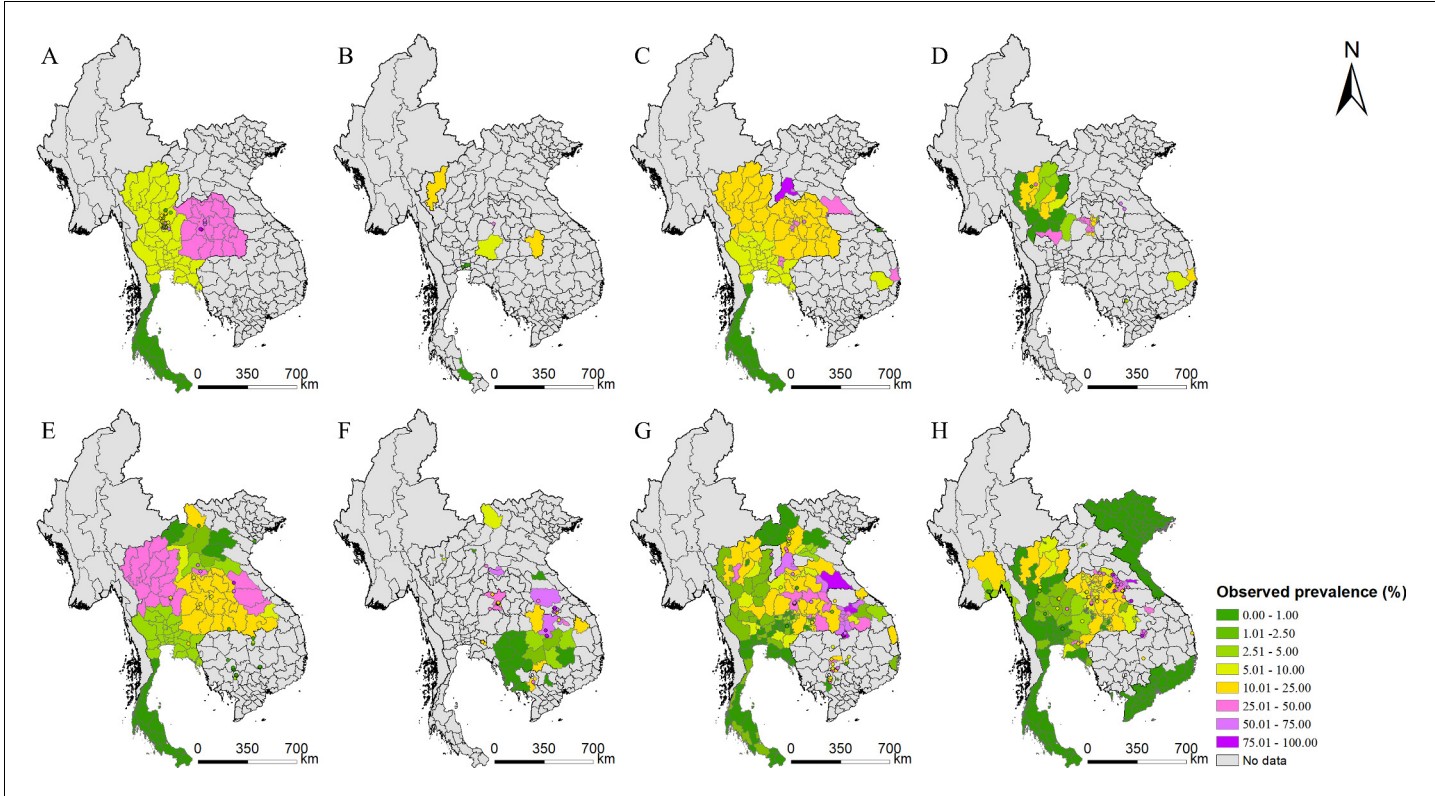

**Figure 2.** Survey locations and observed prevalence of *O. viverrini* infection in endemic countries of Southeast Asia. (**A**) 1978–1982, (**B**) 1983–1987, (**C**) 1988–1992, (**D**) 1993–1997, (**E**) 1998–2002, (**F**) 2003–2007, (**G**) 2008–2012, and (**H**) 2013–2018.

The online version of this article includes the following source data and figure supplement(s) for figure 2:

**Source data 1.** The original data of *O. viverrini* infection in endemic countries of Southeast Asia.
**Source data 2.** The results of the preferential sampling test.
**Figure supplement 1.** Result of quality assessment of eligible studies.
**Figure supplement 1—source data 1.** The results of quality assessment.

in the logit of the prevalence. And increase in 1 m in elevation was associated with the 0.003 (95% BCI: 0.001–0.005) decrease in the logit of the prevalence. The spatial range was estimated as 83.55 km (95% BCI: 81.34–86.61), the spatial variance $\sigma^2_\phi$ was 12.59 (95% BCI: 11.96–13.56), the variance of beta-likelihood $\sigma^2_\beta$ was 0.15 (95% BCI: 0.14–0.15), and the temporal correlation coefficient $\rho$ was 0.66 (95% BCI: 0.65–0.67). Model validation showed that our model was able to correctly estimate 79.61% of locations within the 95% BCI, indicating the model had a reasonable capacity of prediction accuracy. The ME, MAE, and MSE were 0.24%, 9.06%, and 2.38%, respectively, in the final model, while they were −7.14%, 16.67%, and 5.09%, respectively, in the model only based on point-referenced data, suggesting that the performance of the final model was better than the model only based on point-referenced data. On the other hand, Monte Carlo test for preferential sampling suggested that preferential sampling may exist for survey locations in one third (6/18) of the survey years (*Figure 2—source data 2*).

The estimated risk maps of *O. viverrini* infection in different selected years (i.e., 1978, 1983, 1988, 1993, 1998, 2003, 2008, 2013, and 2018) are presented in *Figure 3*. In 2018, the high infection risk (with prevalence >25%) was mainly estimated in regions of the southern, the central, and the north-central parts of Lao PDR, some areas in the east-central parts of Cambodia, and some areas of the northeastern and the northern parts of Thailand. The southern part of Thailand, the northern part of Lao PDR, and the western part of Cambodia showed low risk estimates (with prevalence <5%) of *O. viverrini* infection. The central and several southern parts of Vietnam showed low to moderate risk of *O. viverrini* infection, while there was no evidence of *O. viverrini* in other parts of Vietnam. High

**Table 2.** Posterior summaries of model parameters.

| | Estimated median (95% BCI) | OR | Prob (%)* |
|---|---|---|---|
| Intercept | −4.51 (−5.08, −3.94) | | |
| Survey type | | | |
| School-based survey | Ref | Ref | - |
| Community-based survey | 0.96 (0.70, 1.23) | 2.61 (2.10, 3.42) | >99.99 |
| Diagnostic methods | | | |
| Kato–Katz | Ref | Ref | - |
| FECT | −0.28 (−0.49, −0.07) | 0.76 (0.61, 0.93) | 0.80 |
| Other methods | 0.01 (−0.07, 0.10) | 1.01 (0.93, 1.12) | 64.20 |
| Land surface temperature (LST) in the daytime (°C) | | | |
| <30.65 | Ref | Ref | - |
| 30.65–32.07 | 0.25 (−0.001, 0.50) | 1.28 (0.999, 1.65) | 97.40 |
| >32.07 | 0.07 (−0.18, 0.33) | 1.07 (0.84, 1.39) | 73.40 |
| Human influence index | −0.01 (−0.02, −0.003) | 0.99 (0.98, 1.00) | 0.80 |
| Distance to the nearest open water bodies (km) | 0.24 (−1.45, 1.94) | 1.27 (0.23, 6.96) | 60.20 |
| Elevation (m) | −0.003 (−0.005,−0.001) | 0.997 (0.995, 0.999) | <0.01 |
| Travel time to the nearest big city (min) | 0.0001 (−0.002, 0.002) | 1.00 (0.998, 1.002) | 56.60 |

*Posterior probability of OR > 1.

estimation uncertainty was mainly present in the central part of Lao PDR, the northern and the eastern parts of Thailand, and the central part of Cambodia and Vietnam (*Figure 4*).

In addition, the infection risk varies over time across the study region (*Figure 5*). Areas of northern Thailand showed an increasing trend in periods 1978–1988 and 1993–2003, while most areas of the country presented a considerable decrease of infection risk after 2008. The infection risk first increased and then decreased in areas of the north, the central, and the southern parts of Lao PDR and the central parts of Vietnam. The east-central and western part of Cambodia showed an increasing trend in recent years.

The population-adjusted estimated prevalence over the study region presents a trend down after 1995 (*Figure 6* and *Figure 6—figure supplements 1–9*). At the country level, the estimated prevalence in Thailand showed a fast decline after 1995 and took on a gradually decreasing change in Cambodia. In Lao PDR, the overall prevalence maintained quite stable before 1990 and decreased slightly between 1990 and 1997, increased significantly after 1997, then decreased from 2006, and became stable after 2011. The prevalence is stable in Vietnam during the whole study period. We estimated that in 2018, the overall population-adjusted estimated prevalence of *O. viverrini* infection in the whole study region was 6.57% (95% BCI: 5.35–7.99%), corresponding to 12.39 million (95% BCI: 10.10–15.06) infected individuals (*Table 3*). Lao PDR showed the highest prevalence (35.21%, 95% BCI: 28.50–40.70%), followed by Thailand (9.71%, 95% BCI: 7.98–12.17%), Cambodia (6.15%, 95% BCI: 2.41–11.73%), and Vietnam (2.15%, 95% BCI: 0.73–4.40%). Thailand had the largest numbers of individuals estimated to be infected with *O. viverrini* (6.71 million, 95% BCI: 5.51–8.41), followed by Lao PDR (2.45 million, 95% BCI: 1.98–2.83), Vietnam (2.07 million, 95% BCI: 0.70–4.24), and Cambodia (1.00 million, 95% BCI: 0.39–1.90).

## Discussion

In this study, we produced model-based, high-resolution risk estimates of opisthorchiasis across endemic countries of Southeast Asia. The disease is the most important foodborne trematodiasis in the study region (*Sripa et al., 2010*), taking into account most of the disease burden of opisthorchiasis in the world (*Fürst et al., 2012*). The estimates were obtained by systematically reviewing all possible geo-referenced survey data and applying a Bayesian geostatistical modeling approach that jointly analyzes point-referenced and area-aggregated disease data, as well as environmental and

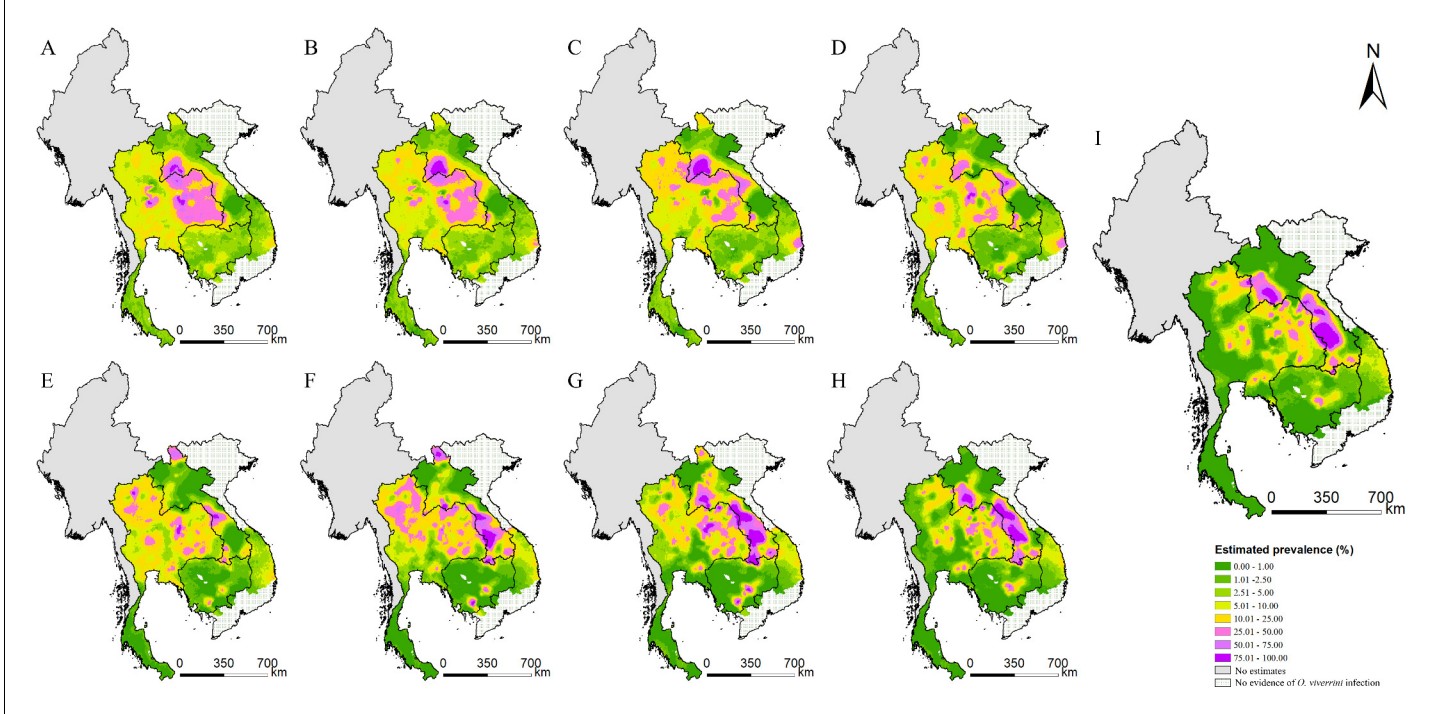

**Figure 3.** Model-based estimated risk maps of *O. viverrini* infection in endemic countries of Southeast Asia in different years. Estimated prevalence based on the median of the posterior estimated distribution of infection risk in (A) 1978, (B) 1983, (C) 1988, (D) 1993, (E) 1998, (F) 2003, (G) 2008, (H) 2013, and (I) 2018.

The online version of this article includes the following source data and figure supplement(s) for figure 3:

**Source data 1.** The sensitivity analysis results of model-based estimated risk maps in 2018.

**Figure supplement 1.** Model-based estimated risk maps of *O. viverrini* infection in 2018 under different values assigned to prevalence for surveys only reported prevalence in intervals.

**Figure supplement 1—source data 1.** Sensitivity analysis for surveys reported prevalence in intervals.

socioeconomic predictors. Our findings will be important for guiding control and intervention cost-effectively and serve as a baseline for future progress assessment.

Our estimates suggested that there was an overall decrease of *O. viverrini* infection in Southeast Asia from 1995 onwards, which may be largely attributed to the decline of infection prevalence in Thailand. This decline was probably on account of the national opisthorchiasis control program launched by the Ministry of Public Health of Thailand from 1987 (*Jongsuksuntigul and Imsomboon, 2003*; *Jongsuksuntigul et al., 2003*). Our high-resolution risk estimates in Thailand in 2018 showed similar pattern as the climatic suitability map provided by Suwannatrai and colleagues (*Suwannatrai et al., 2017*). In this case, we estimated the prevalence of the population instead of the occurrence probability of the parasite, which arms decision makers with more direct epidemiological information for guiding control and intervention. The national surveys in Thailand reported a prevalence of 8.7% and 5.2% in 2009 and 2014, respectively (*MOPH, 2014*; *Wongsaroj et al., 2014*). However, we estimated higher prevalence of 12.44% (95% BCI: 10.79–14.26%) and 9.34% (95% BCI: 7.88–11.02%) in 2009 and 2014, respectively. Even though the national surveys covered most provinces in Thailand, estimates were based on simply calculating the percentage of positive cases among all the participants (*Wongsaroj et al., 2014*), and the remote areas might not be included (*Maipanich et al., 2004*). Instead, our estimates were based on rigorous Bayesian geostatistical modeling of available survey data with environmental and socioeconomic predictors, accounting for heterogeneous distribution of infection risk and population density when aggregating country-level prevalence.

Our findings suggested that the overall prevalence of *O. viverrini* remained high (>20%) in Lao PDR during the study periods, consistent with conclusions drawn by *Suwannatrai et al., 2018*. We

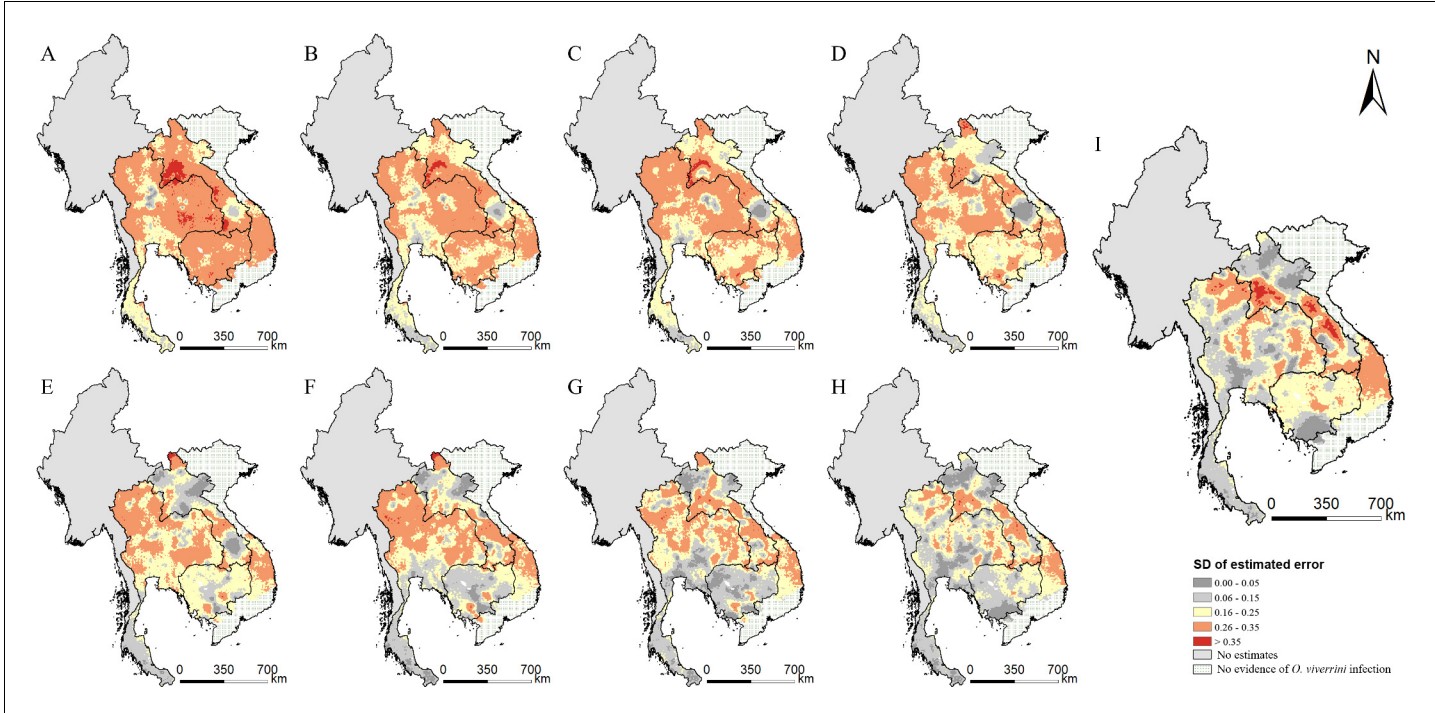

**Figure 4.** The estimation uncertainty in endemic countries of Southeast Asia in different years. (**A**) 1978, (**B**) 1983, (**C**) 1988, (**D**) 1993, (**E**) 1998, (**F**) 2003, (**G**) 2008, (**H**) 2013, and (**I**) 2018.

The online version of this article includes the following source data for figure 4:

**Source data 1.** The results of the estimated uncertainty in endemic countries of Southeast Asia in different years.

estimated that a total number of 2.45 (95% BCI: 1.98–2.83) million people living in Lao PDR were infected with *O. viverrini*, equivalent to that estimated by WHO in 2004 (*WHO, 2002*). Besides, our risk mapping for Champasack province shares similarly risk map pattern produced by Forrer and colleagues (*Forrer et al., 2012*). A national-scaled survey in Cambodia during the period 2006–2011 reported infection rate of 5.7% (*Yong et al., 2014*), lower than our estimation of 8.34% (95% BCI: 5.25–14.95%) in 2011. The former may underestimate the prevalence because more than 77% of participants were schoolchildren (*Yong et al., 2014*). Another large survey in five provinces of Cambodia suggested a large intra-district variation, which makes the identification of endemic areas difficult (*Miyamoto et al., 2014*). Our high-resolution estimates for Cambodia help to differentiate the intra-district risk. However, the estimates should be taken cautious due to large district-wide variances and a relatively small number of surveys. Indeed, *O. viverrini* infection was underreported in Cambodia (*Khieu et al., 2019*), and further point-referenced survey data are recommended for more confirmative results.

Although an overall low prevalence was estimated in Vietnam (2.15%, 95% BCI: 0.73–4.40%) in 2018, it corresponds to 2.07 million (95% BCI: 0.70–4.24 million) people infected, comparable to the number in Lao PDR, mainly due to a larger population in Vietnam. The risk mapping suggested moderate to high risk areas presented in central Vietnam, with a high risk in Phu Yen province for many years, particularly. This agreed with previous studies considering the province a 'hotspot' (*Doanh and Nawa, 2016*). Of note, even though there was no evidence of *O. viverrini* infection in the northern part of the country, *Clonorchis sinensis*, another important liver fluke species, is endemic in the region (*Sithithaworn et al., 2012*). We did not provide estimates for Myanmar in case of large estimated errors. Indeed, only two relevant papers were identified by our systematic review, where one shows low to moderate prevalence in three regions of Lower Myanmar (*Aung et al., 2017*), and the other found low endemic of *O. viverrini* infection in three districts of the capital city Yangon (*Sohn et al., 2019*). Nation-wide epidemiological studies are urgent for a more comprehensive understanding of the disease in Myanmar.

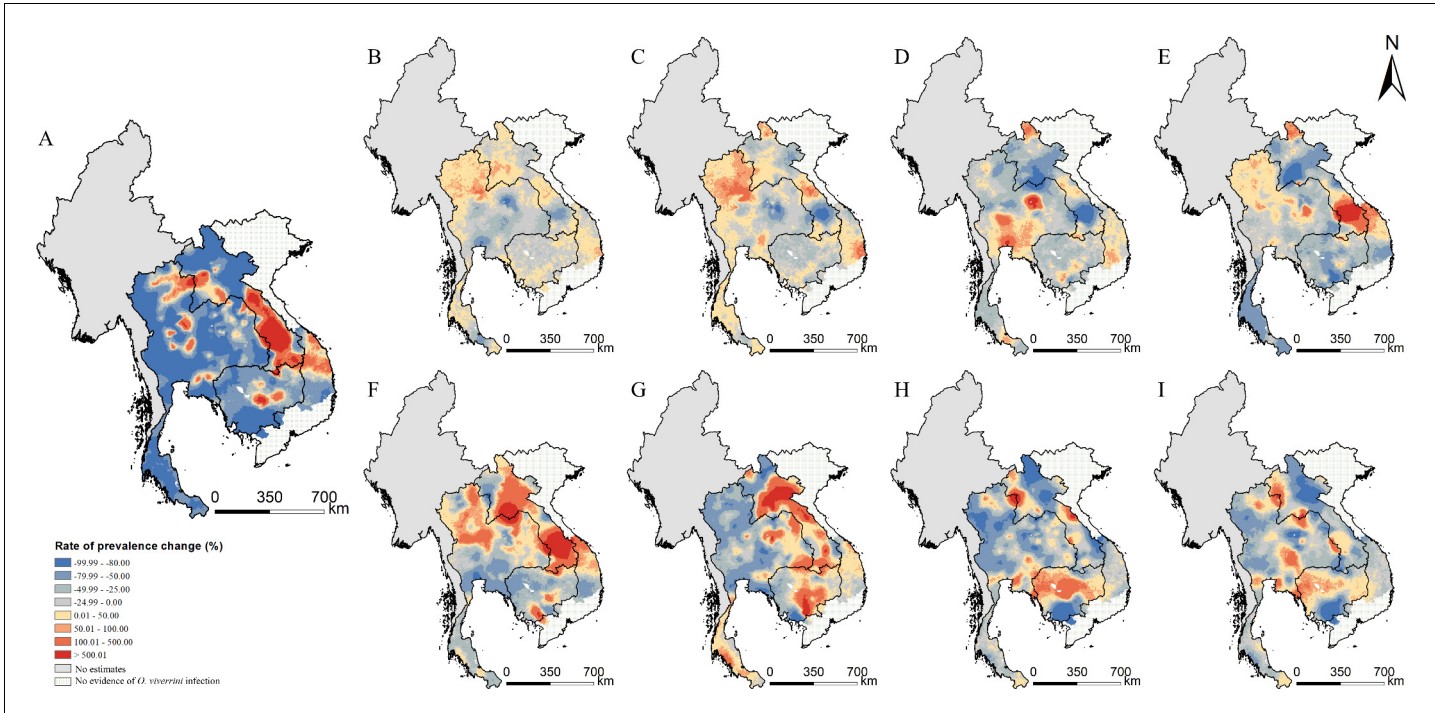

**Figure 5.** Changes of *O. viverrini* infection risk across time periods. Changes were calculated by the median of the posterior estimated distribution of infection risk for the latter time period minus that for the former time period divided by that for the former time period. The risk changes (**A**) between 1978 and 2018; (**B**) between 1978 and 1983; (**C**) between 1983 and 1988; (**D**) between 1988 and 1993; (**E**) between 1993 and 1998; (**F**) between 1998 and 2003; (**G**) between 2003 and 2008; (**H**) between 2008 and 2013; and (**I**) between 2013 and 2018 (source data: *Figure 5—source data 1*).

The online version of this article includes the following source data for figure 5:

**Source data 1.** The results of the changes of *O. viverrini* infection risk across time periods.

We identified several important factors associated with *O. viverrini* infection in Southeast Asia, which may provide insights for the prevention and control of the disease. The infection risk was higher in the entire community than that in schoolchildren, consistent with multiple studies (*Aung et al., 2017*; *Forrer et al., 2012*; *Miyamoto et al., 2014*; *Van De, 2004*; *Wongsaroj et al., 2014*). A negative association was found between *O. viverrini* infection and elevation, suggesting the disease was more likely to occur in low altitude areas, which was consistent with a previous study (*Wang et al., 2013*). HII, a measure of human direct influence on ecosystems (*Sanderson et al., 2002*), showed a negative relationship with *O. viverrini* infection risk, indicating the disease was more likely to occur in areas with low levels of human activities, which were often remote and economically underdeveloped. The habit of eating raw or insufficiently cooked fish was more common in rural areas than that in economically developed ones, which could partially explain our findings (*Grundy-Warr et al., 2012*, *Keiser, 2019*). Indeed, this culturally rooted habit is one of the determinants for human opisthorchiasis (*Kaewpitoon et al., 2008*; *Ziegler et al., 2011*). However, the precise geographical distribution of such information is unavailable and thus we could not use it as a covariate in this study.

Our estimate of the number of people infected with *O. viverrini* is higher than that of the previous study (12.39 million vs 8.6 million [*Qian and Zhou, 2019*]) emphasizing the public health importance of this neglected disease in Southeast Asia, and suggesting that more effective control interventions should be conducted, particularly in the high risk areas. The successful experience in the intervention of Thailand may be useful for reference by other endemic countries of the region. The national opisthorchiasis control program, supported by the government of Thailand, applied interrelated approaches, including stool examination and treatment of positive cases, health education aiming at the promotion of cooked fish consumption, and environmental sanitation to improve hygienic defecation (*Jongsuksuntigul and Imsomboon, 2003*). In addition, for areas with difficulties to reduce infection risk, a new strategy was developed by Sripa and colleagues, using the EcoHealth approach

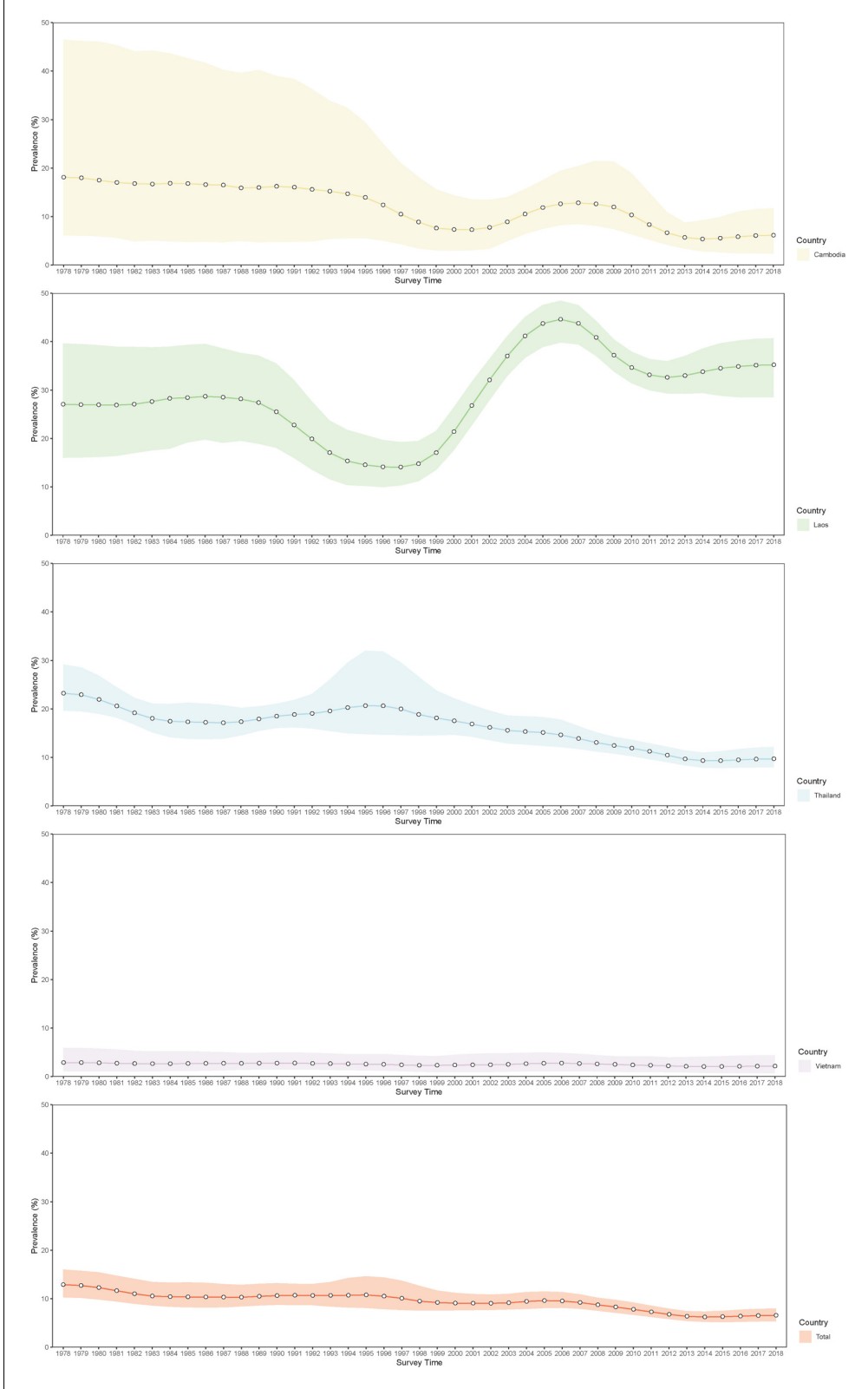

**Figure 6.** Trends in estimated prevalence of *O. viverrini* infection in Southeast Asia.

The online version of this article includes the following source data and figure supplement(s) for figure 6:

**Source data 1.** The results of the estimated prevalence of *O. viverrini* infection in Southeast Asia in different years.

*Figure 6 continued on next page*

*Figure 6 continued*

**Figure supplement 1.** The population-adjusted estimated prevalence (median ± 95% BCI) in 2018 in four countries at administrative division of level 1.

**Figure supplement 1—source data 1.** The results of the population-adjusted estimated prevalence in 2018 in four countries at administrative division of level 1.

**Figure supplement 2.** The population-adjusted estimated prevalence (median ± 95% BCI) in 2013 in four countries at administrative division of level 1.

**Figure supplement 2—source data 1.** The results of the population-adjusted estimated prevalence in 2013 in four countries at administrative division of level 1.

**Figure supplement 3.** The population-adjusted estimated prevalence (median ± 95% BCI) in 2008 in four countries at administrative division of level 1.

**Figure supplement 3—source data 1.** The results of the population-adjusted estimated prevalence in 2008 in four countries at administrative division of level 1.

**Figure supplement 4.** The population-adjusted estimated prevalence (median ± 95% BCI) in 2003 in four countries at administrative division of level 1.

**Figure supplement 4—source data 1.** The results of the population-adjusted estimated prevalence in 2003 in four countries at administrative division of level 1.

**Figure supplement 5.** The population-adjusted estimated prevalence (median ± 95% BCI) in 1998 in four countries at administrative division of level 1.

**Figure supplement 5—source data 1.** The results of the population-adjusted estimated prevalence in 1998 in four countries at administrative division of level 1.

**Figure supplement 6.** The population-adjusted estimated prevalence (median ± 95% BCI) in 1993 in four countries at administrative division of level 1.

**Figure supplement 6—source data 1.** The results of the population-adjusted estimated prevalence in 1993 in four countries at administrative division of level 1.

**Figure supplement 7.** The population-adjusted estimated prevalence (median ± 95% BCI) in 1988 in four countries at administrative division of level 1.

**Figure supplement 7—source data 1.** The results of the population-adjusted estimated prevalence in 1988 in four countries at administrative division of level 1.

**Figure supplement 8.** The population-adjusted estimated prevalence (median ± 95% BCI) in 1983 in four countries at administrative division of level 1.

**Figure supplement 8—source data 1.** The results of the population-adjusted estimated prevalence in 1983 in four countries at administrative division of level 1.

**Figure supplement 9.** The population-adjusted estimated prevalence (median ± 95% BCI) in 1978 in four countries at administrative division of level 1.

**Figure supplement 9—source data 1.** The results of the population-adjusted estimated prevalence in 1978 in four countries at administrative division of level 1.

with anthelminthic treatment, novel intensive health education on both communities and schools, ecosystem monitoring, and active participation of the community (*Sripa et al., 2015*). This 'Lawa model' shows good effectiveness in Lawa Lake area, where the liver fluke was highly endemic (*Sripa et al., 2015*). Furthermore, common integrated control interventions (e.g., combination of preventive chemotherapy with praziquantel, improvement of sanitation and water sources, and health education) are applicable not only for opisthorchiasis but also for other NTDs, such as soil-transmitted helminth infection and schistosomiasis, which are also prevalent in the study region (*Dunn et al., 2016*; *Gordon et al., 2019*). Implementation of such interventions in co-endemic areas could be cost-effective (*Linehan et al., 2011*; *WHO, 2012*).

Frankly, several limitations exist in our study. We collected data from different sources, locations of which might not be random and preferential sampling may exist. We performed a risk-preferential sampling test and the results showed that preferential sampling might exist for survey locations in one third (6/18) of the survey years (*Figure 2—source data 2*). The corresponding impacts might

**Table 3.** Population-adjusted estimated prevalence and number of individuals infected with *O. viverrini* in endemic countries of Southeast Asia in 2018[*].

|  | Population ($\times 10^3$) | Prevalence (%) | No. infected ($\times 10^3$) |
|---|---|---|---|
| Cambodia | 16227.39 | 6.15 (2.41, 11.73) | 997.95 (390.46, 1903.46) |
| Lao PDR | 6960.28 | 35.21 (28.50, 40.70) | 2450.54 (1983.38, 2832.96) |
| Thailand | 69112.64 | 9.71 (7.98, 12.17) | 6708.68 (5514.87, 8411.98) |
| Vietnam | 96421.69 | 2.15 (0.73, 4.40) | 2073.72 (703.46, 4244.85) |
| Total | 188722.01 | 6.57 (5.35, 7.98) | 12389.69 (10099.29, 15060.18) |

[*]Estimates were based on gridded population of 2018 and the median and 95% BCI of the posterior estimated distribution of the infection risk in 2018.

include improper variogram estimator, biased parameter estimation, and unreliable exposure surface estimates (*Diggle et al., 2010*; *Pati et al., 2011*; *Gelfand et al., 2012*). To avoid a more complex model, we did not take into account the preferential sampling issue for our final model, as the model validation showed a reasonable capacity of prediction accuracy. However, the disadvantage of this issue should be well aware.

We set clear criteria for selection of all possible qualified surveys and did not exclude surveys that reported prevalence in intervals without exact observed values. Sensitivity analysis showed that the using the midpoint values of the intervals had little effects on the final results (*Figure 3—figure supplement 1*). For surveys across a large area, complex designs, such as randomly sampling from subgroups of the population under a well-designed scheme, are likely adopted, as it is impractical to draw simple random samples from the whole area. In such case, respondents may have unequal probabilities to be selected, thus weighting should be used to generalize results for the entire area. The observed disease data we collected were from surveys either at point-level (i.e., community or school) or aggregated over areas. For point-level data, as study areas were quite small, simple sampling design was mostly used in the corresponding surveys. And for areal-level data, particularly those aggregated across ADM1, complex designs were likely applied. However, most of the corresponding surveys were only reported raw prevalence or prevalence without clarifying whether weighting was applied. Thus, we did not have enough information to address the design effect for each single survey included. On the other hand, as population density across the study region was different, we calculated the estimated country- and provincial-level prevalence by averaging the estimated pixel-level prevalence weighted by population density. In this way, we took into account the diversity of population density across areas for regional summaries of the estimates.

We assumed similar proportions of age and gender in different surveys, as most of which only reported prevalence aggregated by age and gender. Nevertheless, considering the possible differences in infection risk between the whole population and schoolchildren, we categorized survey types to the community- and school-based. Furthermore, our analysis was based on survey data under different diagnostic methods. The sensitivity and specificity of the same diagnostic method may differ across studies (*Charoensuk et al., 2019*; *Laoprom et al., 2016*; *Sayasone et al., 2015*), while different diagnostic methods may result in different results in the same survey. To partially taking into account the diversity of diagnostic methods, we assumed the same diagnostic method has similar sensitivity and specificity, and we considered the types of diagnostic methods as covariates in the model. Results showed that the odds of infection with FECT methods was significantly lower than that with Kato–Katz, which was consistent with results found by *Lovis et al., 2009*. In addition, most of the diagnostic methods in the surveys were based on fecal microscopic technique on eggs, which could not effectively distinguish between *O. viverrini* and minute intestinal flukes of the family Heterophyidae (e.g., heterophyid and lecithodendriid) (*Charoensuk et al., 2019*, *Sato et al., 2010*). Thus, our results may overestimate the *O. viverrini* infection risk in areas where heterophyid and lecithodendriid are endemic, such as Phongsaly, Saravane, and Champasak provinces in Lao PDR (*Sato et al., 2010*, *Chai et al., 2010*; *Chai et al., 2013*), Nan and Lampang provinces in Thailand (*Wijit et al., 2013*), and Takeo province in Cambodia (*Sohn et al., 2011*). There is an urgent need for the application of more powerful diagnostic practices with higher sensitivity and specificity to better detect the true O. viverrini prevalence, such as PCR (*Lovis et al., 2009*, *Lu et al., 2017*, *Sato et al., 2010*). Nevertheless, because of the similar treatment and the prevention strategies of *O. viverrini* and minute intestinal flukes (*Keiser and Utzinger, 2010*), our risk mapping is valuable also for areas co-endemic with the above flukes.

In conclusion, this study contributes to better understand the spatial-temporal characteristics of *O. viverrini* infection in major endemic countries of Southeast Asia, providing valuable information guiding control and intervention, and serving as a baseline for future progress assessment. Estimates were based on a rigorous geostatistical framework jointly analyzing point- and areal-level survey data with potential predictors. The higher number of infected people we estimated highlights the public health importance of this neglected disease in the study region. More comprehensive epidemiological studies are urgently needed for endemic areas with scant survey data.

# Materials and methods

### Key resources table

| Reagent type (species) or resource | Designation | Source or reference | Identifiers | Additional information |
|---|---|---|---|---|
| Software, algorithm | R Project for Statistical Computing | R Project for Statistical Computing | RRID:SCR_001905 | |
| Software, algorithm | ArcGIS for Desktop Basic | ArcGIS for Desktop Basic | RRID:SCR_011081 | |
| Software, algorithm | R-INLA Project | R-INLA Project | | https://www.r-inla.org/ |
| Software, algorithm | 'PStestR' R Package | 'PStestR' R Package | | https://github.com/joeno middlename/PStestR |

## Search strategy, selection criteria, and data extraction

We collected relevant publications reporting prevalence data of opisthorchiasis in Southeast Asia through a systematic review (registered in the International Prospective Register of Systematic Reviews, PROSPERO, No.CRD42019136281), and reported our systematic review according to the PRISMA guidelines (*Supplementary file 1A*; *Moher et al., 2010*). We searched PubMed and ISI Web of Science from inception to February 9, 2020, with search terms: (liver fluke* OR Opisthorchi*) AND (Southeast Asia OR Indonesia OR (Myanmar OR Burma) OR Thailand OR Vietnam OR Malaysia OR Philippines OR Lao PDR OR Cambodia OR Timor OR Brunei OR Singapore). We set no limitations on language, date of survey, or study design in our search strategy. For literatures not found by the above methods, we also reviewed reports from governments or Ministry of Health, theses, relevant books, and documents.

We followed a protocol (*Figure 1—figure supplement 1*) for inclusion, exclusion, and extraction of survey data. First, we screened titles and abstracts to identify potentially relevant articles. Publications on in vitro studies, or absence of human studies or absence of disease studies were excluded. Quality control was undertaken by re-checking 20% of randomly selected irrelevant papers. Second, the full-text review was applied to potentially relevant articles. We excluded publications with following conditions: absence of prevalence data; studies done in specific patient groups (e.g., prevalence on patients with specific diseases), in specific population groups (e.g., travelers, military personnel, expatriates, nomads, displaced or migrating population), under specific study designs (e.g., case report studies, case–control studies, clinical trials, autopsy studies); drug efficacy or intervention studies (except for baseline data or control groups), population deworming within 1 year, the survey time interval more than 10 years, data only based on the direct smear method (due to low sensitivity) or serum diagnostics (due to unable to differ the past and the active infection). During the full-text review, the potential relevant cited references of the articles were also screened. Studies were included if they reported survey data at provincial level and below, such as administrative divisions of level 1 (ADM1: province, state, etc.), 2 (ADM2: city, etc.), and 3 (ADM3: county, etc.), and at point-level (village, town, school, etc.). Duplicates were checked and removed. The quality assessment of each individual record included in the final geostatistical analysis was performed by two independent reviewers, based on a nine-point quality evaluation checklist (*Figure 2—figure supplement 1—source data 1*).

We followed the GATHER checklist (*Supplementary file 1B*; *Stevens et al., 2016*) for the data extraction. Detailed information of records was extracted into a database, which includes literature information (e.g., journal, authors, publication date, title, volume, and issue), survey information (e.g., survey type: community- or school-based, and year of survey), location information (e.g., location name, location type, and coordinates), and disease-related data (e.g., species of parasites, diagnostic method, population age, number of examined, number of positive, and percentage of positive). The coordinates of the survey locations were obtained from Google Maps (https://www.google.com/maps/). For surveys reported prevalence in intervals without exact observed values, the midpoints of the intervals were assigned.

### Environmental, socioeconomic, and demographic data

The environmental data (i.e., annual precipitation, distance to the nearest open water bodies, elevation, land cover, land surface temperature [LST] in the daytime and at night, and normalized difference vegetation index [NDVI]), socioeconomic data (i.e., human influence index, survey type, and travel time to the nearest big city), and demographic data of Southeast Asia were downloaded from open data sources (*Figure 7—source data 1*). Land cover data was summarized by the most frequent category within each pixel over the period of 2001–2018. We combined similar land cover classes and re-grouped them into five categories: (i) croplands; (ii) forests; (iii) shrub and grass; (iv) urban; and (v) others. LST in the daytime and at night, as well as NDVI were averaged over the period of 2000–2018. All data were aligned over a 5 × 5 km grid across the study region (*Figure 7*). Data at point-referenced survey locations were extracted. We linked the data to the divisions (i.e., ADM1, ADM2, or ADM3) reported aggregated outcome of interest (i.e., infection prevalence) by averaging them within the corresponding divisions. The above data processing was done using the package 'ratser' (https://cran.r-project.org/web/packages/raster) through R (version 3.5.0).

### Model fitting and variable selection

As our outcome of interest derived from both point-referenced and area-aggregated surveys, a bivariate Bayesian geostatistical joint modeling approach was applied to analyze the area-level and point-level survey data together (*Moraga et al., 2017*; *Utazi et al., 2019*), and account for both disease data reporting numbers of examined and positive, and those reporting only prevalence.

We defined $p_{it}$ the probability of infection at location $i$ and time period $t$, where $i$ is the index either for the location of point-referenced data or of the area for area-level data. Based on the probability theory, for data reported with numbers of examined and positive, we assumed that the number of examined $Y_{it}$ followed a binomial distribution $Y_{it} \sim Bin(p_{it}, N_{it})$, where $N_{it}$ denoted the number of examined; and for data only reported with the observed prevalence, we assumed that the

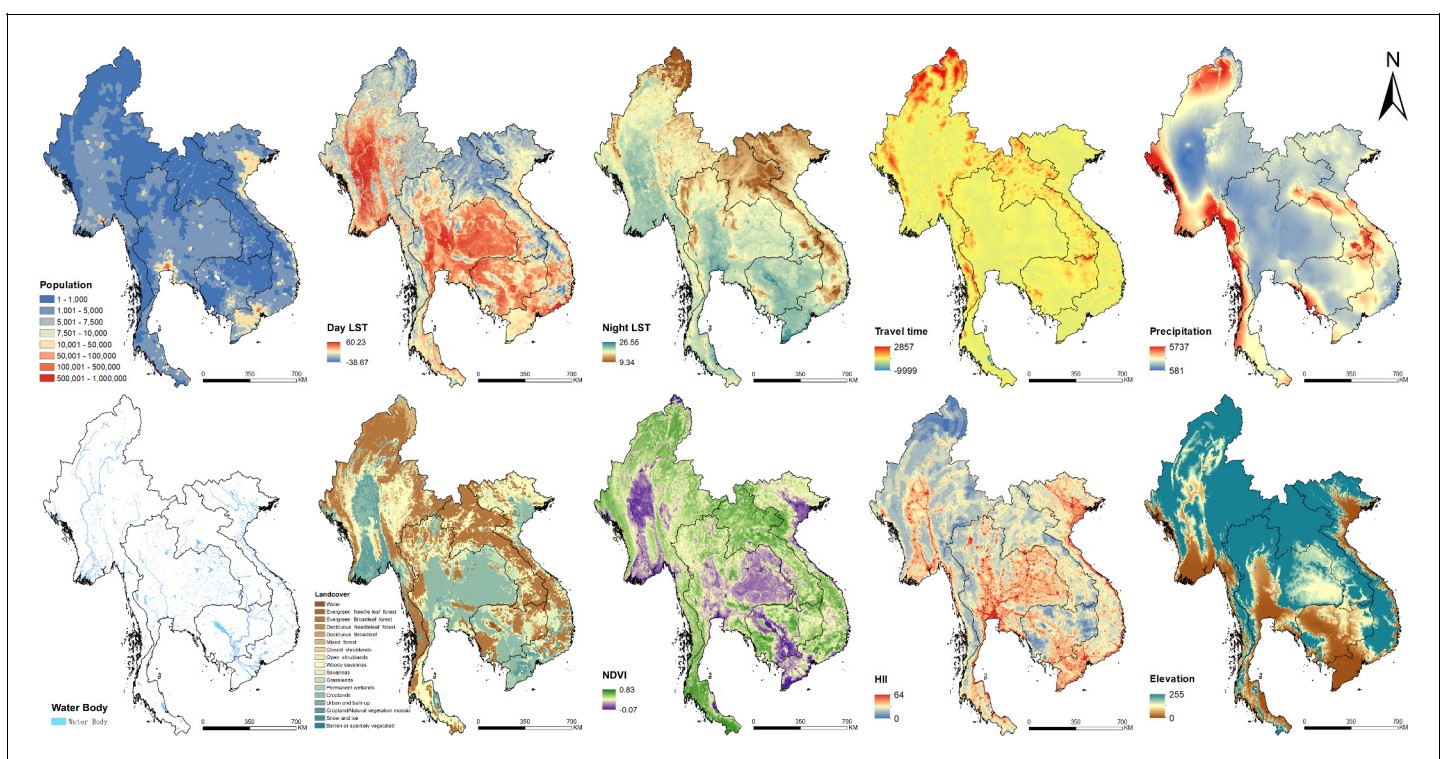

**Figure 7.** Images of spatial covariates used in the present study.

The online version of this article includes the following source data for figure 7:

**Source data 1.** The sources of covariate layers.

observed prevalence $ob_{it}$ followed a beta distribution $ob_{it} \sim Be\left(p_{it}, \sigma_\beta^2\right)$. The period of this study was from 1978 to 2018. We modeled predictors on a logit scale of $p_{it}$.

We referred to the method proposed by Cameletti and colleagues (*Krainski, 2019*; *Cameletti et al., 2013*) to build a spatial-temporal model combined with covariates, which was defined as an SPDE (Stochastic Partial Differential Equation) model for the spatial domain and an AR1 model for the time dimension. A standard grid of $5 \times 5$ km$^2$ was overlaid to each survey area resulting in a certain number of pixels representing the area. We assumed that survey locations and pixels within survey areas shared the same spatial-temporal process. In addition, we assumed the infection risk the same within 1-year period for the same areas. Different observations from the same year in the same areas can be treated as realizations of the randomized spatial-temporal process. Let $i = 1, \dots, n_A, n_A + 1, \dots, n_A + n_p$, where $n_A$ is the total number of areas for area-level surveys and $n_p$ is the total number of locations for point-referenced surveys. Regarding area-level data, $\text{logit}(p_{it}) = \beta_0 + \tilde{\bm{x}}'_{it}\beta + |A_i|^{-1}\int_{A_i} \omega(s,t)dsdt$, where $i = 1, \dots, n_A$, $\tilde{\bm{x}}_{it}$ the vectors of covariate values for $i^{th}$ area in time period $t$ with $\tilde{\bm{x}}'_{it} = |A_i|^{-1}\int_{A_i} x(s,t)dsdt$ and $\beta_0$ and $\beta$ are the intercept and the corresponding regression coefficients. $|A_i| = \int_{A_i} 1ds$ is the size of the $i^{th}$ area and $\omega(s,t)$ the spatial-temporal random effects of pixels within the area. For point-referenced data, $\text{logit}(p_{it}) = \beta_0 + \bm{x}'_{it}\beta + \omega(s_i, t)$, where $i = n_A + 1, \dots, n_A + n_p$, $\bm{x}'_{it}$ is the vectors of covariate values and $\omega(s_i, t)$ is the spatial-temporal random effect for $i^{th}$ location in time period $t$. To decrease the computational burden, under the SPDE framework, we built the GMRF on regular temporal knots, that is, $\omega = (\omega_{t=1978}, \omega_{t=1983}, \omega_{t=1988}, \omega_{t=1993}, \omega_{t=1998}, \omega_{t=2003}, \omega_{t=2008}, \omega_{t=2013}, \omega_{t=2018})'$ (*Cameletti et al., 2013*; *Krainski, 2019*). We assumed the spatio-temporal random effect $\omega(s,t)$ follow a zero-mean Gaussian distribution, that is, $\omega \sim GP(0, \bm{K}_{space} \otimes \bm{K}_{time})$, where the spatial covariance matrix $\bm{K}_{space}$ was defined as a stationary Matérn covariance function $\sigma_\phi^2(\kappa\bm{D})^\nu K_\nu(\kappa\bm{D})/(\Gamma(\nu)2^{\nu-1})$ and the temporal covariance matrix as $\bm{K}_{time} = \rho^{|t_u - t_o|}$ with $|\rho| < 1$, corresponding to the autoregressive stochastic process with first order (AR1). And the spatio-temporal random effect $\omega(s,t)$ was assumed independent of each other in different times and locations, that is, $\text{Cov}\left(\omega_{it}, \omega_{jt'}\right) = \begin{cases} 0, & \text{if } t \neq t' \\ \sigma_\phi^2, & \text{if } t = t' \end{cases}$. Here $\bm{D}$ donates the Euclidean distance matrix, $\kappa$ is a scaling parameter, and the range $r = \sqrt{8\nu}/\kappa$, representing the distance at which spatial correlation becomes negligible (<0.1), and $K_\nu$ is the modified Bessel function of the second kind, with the smoothness parameter $\nu$ fixed at 1. The latent fields corresponding to other years are approximated by projection of $\omega$ using the B-spline basis function of degree two, that is, $B_{i,1}(t) = \begin{cases} 1, & t_i \leq t < t_{i+1} \\ 0, & otherwise \end{cases}$ and $B_{i,m}(t) = \frac{t-t_i}{t_{i+m-1}-t_i}B_{i,m-1}(t) + \frac{t_{i+m}-t}{t_{i+m}-t_{i+1}}B_{i+1,m-1}(t)$, where $m$ is the degree of two (*Krainski, 2019*; *Cameletti et al., 2013*).

We formulated the model in a Bayesian framework. Minimally informative priors were specified for parameters and hyper parameters as follows: $\beta \sim N(0, 10^5\bm{I})$, $\log\left(1/\sigma_\beta^2\right) \sim \log\text{Gamma}(1, 0.1)$, $\log\left(1/\sigma_\phi^2\right) \sim \log\text{Gamma}(1, 0.01)$, $\log((1+\rho)/(1-\rho)) \sim N(0, 0.15)$, and $\log(\kappa) \sim N(\log(\sqrt{8}/d), 1)$, where $d$ is the median distance between the predicted grids.

Additionally, we applied variable selection procedure to identify the best set of predictors for a parsimonious model. First, the best functional form (continuous or categorical) of continuous variables was selected, by fitting univariate Bayesian spatial-temporal models with either form as the independent variable and selecting the form with the lowest log score (*Pettit, 1990*). Second, the best subset method was used to identify the best combination of predictors for the final model. According to previous studies (*Aung et al., 2017*; *Forrer et al., 2012*; *Miyamoto et al., 2014*; *Wongsaroj et al., 2014*), the infection risk in community and school may be different, and using different diagnostic methods may differ the observed prevalence (*Charoensuk et al., 2019*; *Laoprom et al., 2016*; *Sayasone et al., 2015*). Thus, the survey type (i.e., community- or school-based) and the diagnostic methods (i.e., Kato–Katz, FECT, or other methods) were kept in all potential models, while the other 10 environmental and socioeconomic variables were put forth into the Bayesian variable selection process. The model with the minimum log score was chosen as the final model.

Model fitting and variable selection process were conducted through INLA-SPDE approach (**Lindgren et al., 2011**; **Rue et al., 2009**), using INLA package in R (version 3.5.0). Estimation of risk for *O. viverrini* infection in each year of the study period was done over a grid with cell size of $5 \times 5$ km$^2$. And the relative changes of the prevalence were also calculated using a formula as $(pp_{st_j} - pp_{st_i})/pp_{st_i}$ for pixel $s$ between the former year $t_i$ and the later year $t_j$, where $pp$ indicates the median of the posterior estimated distribution of infection risk. The corresponding risk maps and the prevalence changing maps were produced using ArcGIS (version 10.2). In addition, as population density across the study region was different, the population-adjusted estimated prevalence and number of infected individuals in 2018 were calculated at the country and provincial levels averaging the estimated pixel-level prevalence weighted by population density, that is, $pp_A = \sum_{i \in A} pp_i w_i / \sum_{i \in A} w_i$. Here $pp_A$, $pp_i$, and $w_i$ are the estimated prevalence in area $A$, estimated prevalence at pixel $i$, and population density at pixel $i$, respectively, where $i$ belongs to area $A$. Based on previous studies, for the provinces in Vietnam where there was no evidence of *O. viverrini* infection, we multiplied the estimated results by zero as the final estimated prevalence (**Doanh and Nawa, 2016**). The R code used for model fitting is publicly available in GitHub (https://github.com/SYSU-Opisthorchiasis/Spatial-temporal-mapping-of-opisthorchiasis and archived in software heritage; **Zhao, 2021**; copy archived at swh:1:rev:6493df4ba60c1f2f1aaaad979174a3a5d928627a).

## Model validation, sensitivity analysis, and test of preferential sampling

Model validation was conducted using the 5-fold out-of-sample cross-validation approach. Mean error $(\mathrm{ME} = \frac{1}{N} \sum (ob_{it} - pp_{it}))$, mean absolute error $(\mathrm{MAE} = \frac{1}{N} \sum |ob_{it} - pp_{it}|)$, mean square error $(\mathrm{MSE} = \frac{1}{N} \sum (ob_{it} - pp_{it})^2)$, and the coverage rate of observations within 95% BCI were calculated to evaluate the performance of the model. Furthermore, a Bayesian geostatistical model only based on point-referenced data was fitted and validated, to compare its performance with our joint modeling approach. In addition, a sensitivity analysis was conducted to evaluate the effects of using the midpoint values of the intervals as the observed prevalence in one literature from Suwannatrai and colleagues (**Suwannatrai et al., 2018**), reporting observed prevalence of *O. viverrini* infection in intervals. Sensitivity analysis was done by using the lower and the upper limits of the intervals in the modeling analysis.

Considering that the data in this study were sourced from different studies, preferential sampling may exist. We performed a test for preferential sampling of the data. To our knowledge, no method has been developed for preferential sampling test on observations combined at point and areal levels. To compromise, we took centers of the areas with survey data as their locations for the test of preferential sampling. A fast and intuitive Monte Carlo test developed by Watson was adopted for its advantage of fast speed and feasibility of data arising from various distributions. We assumed $S_t$ (i.e., the collection of sampled points at time $t$) a realization from an inhomogeneous Poisson processes (IPP) under the condition of $\omega(s,t)$ (i.e., the spatial-temporal Gaussian random field), that is, $[s_t | \omega(s,t)] = \mathrm{IPP}(\lambda(s,t))$, and $\log(\lambda(s,t)) = \alpha_0 + h(\omega(s,t))$, where $h$ is a monotonic function of $\omega(s,t)$. When $h \equiv 0$, the sampling process is independent from $\omega(s,t)$, thus the preferential sampling is not significant. In this way, the problem of detecting preferential sampling can be transformed into the hypothesis testing of $h \equiv 0$. If $h \equiv 0$ is false, for example, in case that $h$ is a monotonic increasing function of $\omega(s,t)$, then the point patterns $S_t$ are expected to exhibit an excess of clustering in areas with higher $\omega(s,t)$, thus positive association can be detected between the localized amount of clustering and estimated $\omega(s,t)$. First, we used the mean of the distances to the $K$ nearest points ($D_K$) to measure the clustering of locations, and calculated the rank correlation $r_{t(K)}$ between $D_K$ and the estimated $\omega(s,t)$ for survey year $t$. Here the estimated $\omega(s,t)$ was obtained from fitting the Bayesian spatial-temporal joint model. Next, the Monte Carlo method was used to sample realizations from the IPP under the null hypothesis (i.e., $h \equiv 0$), following which a set of rank correlations $r_{t(K)}^M$ were calculated, approximating the distribution of the rank correlations $\rho_{t(K)}$ under $h \equiv 0$. In this way, the nonstandard sampling distribution of the test statistic can be approximated. Finally, we computed the desired empirical p-value by evaluating the proportion of the Monte Carlo-sampled $r_{t(K)}^M$ which are more extreme than $r_{t(K)}$. We set a sample size of 1000 for each Monte Carlo sampling. We also considered $K$ from 1 to 8 to measure the clustering of locations and resulted in eight p-values respective to different $K$ for each survey year. If one of the p-values is smaller or equal to 0.05, we

considered preferential sampling existing in the corresponding survey year. Since our model could estimate the disease risk each year of the study period, this test was done for each survey year with number of locations more than or equal to 10 (i.e., 1978, 1981, 1991, 1995, 1998, 2000, 2001, 2004, 2007, 2008, 2009, 2010, 2011, 2012, 2013, 2014, 2015, and 2016). The test was conducted using the package 'PStestR' in R (version 3.6.3) (*Watson, 2020*).

## Acknowledgements

We are grateful to Dr Roy Burstein from Institute for Disease Modeling, Bellevue, Washington, USA for providing very good suggestions for the manuscript.

## Additional information

### Funding

| Funder | Grant reference number | Author |
| --- | --- | --- |
| National Natural Science Foundation of China | 81703320 | Ying-Si Lai |
| National Natural Science Foundation of China | 82073665 | Ying-Si Lai |
| Natural Science Foundation of Guangdong | 2017A030313704 | Ying-Si Lai |
| China Medical Board | 17-274 | Ying-Si Lai |
| Sun Yat-sen University One Hundred Talent Grant | | Ying-Si Lai |

The funders had no role in study design, data collection and interpretation, or the decision to submit the work for publication.

### Author contributions

Ting-Ting Zhao, Data curation, Formal analysis, Validation, Visualization, Methodology, Writing - original draft, Project administration, Writing - review and editing; Yi-Jing Feng, Data curation, Formal analysis; Pham Ngoc Doanh, Conceptualization, Data curation, Writing - review and editing; Somphou Sayasone, Virak Khieu, Choosak Nithikathkul, Men-Bao Qian, Conceptualization, Writing - review and editing; Yuan-Tao Hao, Conceptualization, Methodology, Writing - review and editing; Ying-Si Lai, Conceptualization, Data curation, Formal analysis, Supervision, Funding acquisition, Validation, Visualization, Methodology, Writing - original draft, Writing - review and editing

### Author ORCIDs

Ting-Ting Zhao (iD) https://orcid.org/0000-0003-2932-2647
Ying-Si Lai (iD) https://orcid.org/0000-0003-4324-5465

### Ethics

Human subjects: This work was based on survey data pertaining to the prevalence of opisthorchiasis extracted from open published peer-reviewed literatures. All data were aggregated and did not contain any information at the individual or household levels. Therefore, there were no specific ethical issues warranted special attention.

### Decision letter and Author response

Decision letter https://doi.org/10.7554/eLife.59755.sa1
Author response https://doi.org/10.7554/eLife.59755.sa2

# Additional files

## Supplementary files

• Supplementary file 1. PRISMA 2009 Checklist and GATHER checklist. Quality assessment: We did quality evaluation for each literature included in the final geostatistical modeling analysis, which is undertaken using a nine-point checklist. The items of quality evaluation are as follows: Q1: provide specific inclusion and exclusion criteria. Q2: provide basic characteristics of the investigated population (gender, age, etc.). Q3: provide prevalence rate of the survey. Q4: provide number of positive patients and number of examined people of the survey. Q5: provide diagnostic method used in the survey. Q6: provide survey type. Q7: provide time of the survey. Q8: describe or discuss the possible bias of the survey or how confounders are controlled. Q9: the literature comes from Science Citation Index Expanded database. Each item is scored 1 in case the publication meets or 0 in contrary. The scores are summed up for all items and assigned to the publication as its quality score. The score for each literature is listed in *Figure 2—figure supplement 1—source data 1*.

• Transparent reporting form

## Data availability

All data generated or analysed during this study are included in the manuscript and supporting files. Source data files have been provided for Figures 2–7, Figure 2-figure supplement 1, Figure 3-figure supplement 1, and Figure 6-figure supplement 1–9.

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
