## [Decision Letter]

**Acceptance summary:**

Infection with *Opithorchis viverrini* is a neglected tropical disease endemic to Southeast Asian countries. This study is particularly interesting in that it compiles data from decades of prevalence surveys of *O. viverrini* infection to produce high resolution maps of infection prevalence in Southeast Asia over the past few decades, identifying regions where prevalence has decreased and increased, and the systemic factors that have influenced the prevalence over time. Such high resolution geographical data will be highly valuable for public health efforts aimed at treating and preventing this disease.

**Decision letter after peer review:**

Thank you for submitting your article "Model-based spatial-temporal mapping of opisthorchiasis in endemic countries of Southeast Asia" for consideration by *eLife*. Your article has been reviewed by three peer reviewers, one of whom is a member of our Board of Reviewing Editors, and, and the evaluation has been overseen by Miles Davenport as the Senior Editor. The reviewers have opted to remain anonymous.

The reviewers have discussed the reviews with one another and the Reviewing Editor has drafted this decision to help you prepare a revised submission.

Summary:

This paper examined the prevalence of opisthorchiasis in Southeast Asian countries using survey data obtained through an extensive literature search. The data comprised both aerial and point-referenced data. The authors then fitted a Bayesian fusion spatiotemporal model to map the prevalence of the disease at 5x5 km resolution. The estimates were also aggregated to different administrative units within the study countries. This work applies state-of-the-art modelling techniques to an interesting application.

Essential revisions:

1) Many of the publications studied only in areas where *Opisthorchis viverrini* is endemic. The prevalence in surveys these are likely to be overestimated due to the preferential sampling of areas. The authors have used a method to test for this (Monte Carlo test using R package PStestR) and claim that they did not detect any preferential sampling. However, the reviewers did not find this very convincing given the clustering of points in Figure 2. We would like the authors to give further details regarding the analysis method they used to test for preferential sampling (Monte Carlo test using R package PStestR), show the results of this analysis, and also to include some further discussion regarding the impact of preferential sampling on the validity of results.

2) The authors stratified predictions by 10-year periods; this is a very coarse time frame for predictions given that incidence of infection can vary from year to year. This limitation should be fully acknowledged by the authors if shorter time frames cannot be used.

3) The reviewers criticized the imputation of sample size in order to convert prevalences to binomial data in papers where sample size was unavailable. While the authors included a sensitivity analysis of the impact of this imputation in Figure 3—source data 3 to help assess this point, this was not considered sufficient to address this issue. The reviewers suggest instead that if the data were originally available as prevalence estimates, these should be treated as such and modelled using a β likelihood or a normal likelihood (on the logit scale) and not converted artificially to binomial data.

4) The authors should describe how they dealt statistically when they encountered multiple estimates from the same area within each of the 10-year periods.

5) Surveys often have complex designs, using weighting to calculate the prevalence over an entire area. How did the authors account for this weighting in their analysis?

6) The authors treated surveys aggregated over ADM2 or ADM3 areas as points, whereas those aggregated over ADM1 areas were treated as areal data. This is a very rough way to handle spatial misalignment. If the data were associated with areas, these should be left as areal data in the analysis and should not be treated as points as one would be enforcing non-existent geographical precision in the data in doing so. The authors should justify this choice, or discuss how it may impact the accuracy of results.

7) The authors used the AUC statistic to validate their model. This is an inappropriate use of the AUC; ROC and AUC are normally used to check the discrimination ability of logistic regression models and not binomial regression models. The authors mention other metrics which are useful for evaluating binomial regression models such as MSE and MAE, but the values of these metrics are not discussed or presented in the manuscript. Please discard the AUC analysis, and instead include a table showing the values of these other metrics in the main manuscript, as well as the bias and 95% coverage rates of the fitted model.

8) The authors discuss differences in test sensitivity as a source of heterogeneity between surveys, which they ignored by assuming similar sensitivity across all surveys. It is unclear how much this may have affected results. Please give estimates of the magnitude of the difference of sensitivity of different diagnostic tests, as this could heavily influence differences in prevalence across surveys if these differences in sensitivity are very large. Is there a reason why the authors did not assess the diagnostic method as a covariate in their model?

9) The authors used as an exclusion criteria surveys using the smear method to detect opisthorchiasis due to its lack of sensitivity. However, in nearly half of all reports, the diagnostic test used was not reported or missing. How do the authors then know that these records did not use the smear method to detect disease?

10) The authors need to provide a list of citations of all their included studies as an appendix, consistent with GATHER item 5 and PRISMA item 18. GATHER also suggests providing a table with each data source used, reference information or contact name/institution, population represented, data collection method, year(s) of data collection, sex and age range, diagnostic criteria or measurement method, and sample size, as relevant.

11) The interpretation of the estimated regression coefficients of the categorical variables was poorly done. In particular for model results Table 2: since the authors used a logit link function, the model results can be converted into odds ratios by exponentiating the model coefficients. Please convert all coefficients in this table into odds ratios. Model coefficients have very little inherent interpretability, while odds ratios can be interpreted by readers as measures of relative risk comparing the reference category and the category in question in relation to the outcome variable. The authors may also want to consider dropping the other non-coefficient model parameters from this table (spatial range, correlation coefficient, spatial variance) and report them in the text instead as their units are not consistent with the rest of the table. For the probability %, this would be reinterpreted as the probability that the odds ratio is >1 for risk factors increasing the prevalence of disease, and <1 for risk factors decreasing the prevalence of disease (distance to nearest open body of water and precipitation). Also, for the variables that were modeled as continuous (precipitation, HII), we need the unit size increase associated with each increase in prevalence (i.e. what increase in annual precipitation is associated with the 0.14 decrease in the logit?)

[Editors' note: further revisions were suggested prior to acceptance, as described below.]

Thank you for resubmitting your work entitled "Model-based spatial-temporal mapping of opisthorchiasis in endemic countries of Southeast Asia" for further consideration by *eLife*. Your revised article has been evaluated by Miles Davenport (Senior Editor) and a Reviewing Editor.

The manuscript has been improved but there are some remaining issues that need to be addressed before acceptance, as outlined below:

1) For Figure 5, negative values are conventionally interpreted as decreases and positive values as increases, so the numbers in this figure are likely to lead to confusion. Please change the calculations instead to (𝑝𝑝𝑠𝑡j − 𝑝𝑝𝑠𝑡i )/𝑝𝑝𝑠𝑡𝑖, which should lead to an inversion of the sign without changing the numbers, and will increase the interpretability of the figure.

2) In Table 2, the exponent of the intercept of the model cannot be interpreted as an odds ratio, as it represents the odds of the prevalence at the reference value of all categories. Please leave the cells for OR and prob(%) blank for this row, as these quantities are not relevant for the intercept.

---

## [Author Response]

Essential revisions:1) Many of the publications studied only in areas where Opisthorchis viverrini is endemic. The prevalence in surveys these are likely to be overestimated due to the preferential sampling of areas. The authors have used a method to test for this (Monte Carlo test using R package PStestR) and claim that they did not detect any preferential sampling. However, the reviewers did not find this very convincing given the clustering of points in Figure 2. We would like the authors to give further details regarding the analysis method they used to test for preferential sampling (Monte Carlo test using R package PStestR), show the results of this analysis, and also to include some further discussion regarding the impact of preferential sampling on the validity of results.

We thank the reviewers for arising the issue of preferential sampling. To our knowledge, there hasn’t been method developed for preferential sampling test on observations combined at point- and areal levels. To compromise, we took centers of the areas with survey data as their locations for the test of preferential sampling. We adopted a fast Monte Carlo test developed by Watson, for its advantage of fast speed and feasibility of data arising from various distributions (Watson, 2020). We assumed St (i.e., the collection of sampled points at time t) a realization from an inhomogeneous Poisson processes (IPP) under the condition of ω(s,t) (i.e., the spatial-temporal Gaussian random field), that is [St|ω(s,t)]=IPP(λ(s,t)), and log(λ(s,t))=α0+h(ω(s,t)), where h is a monotonic function of ω(s,t). When h≡0, the sampling process is independent from ω(s,t), thus the preferential sampling is not significant. In this way, the problem of detecting preferential sampling can be transformed into the hypothesis testing of h≡0. If h≡0 is false, for example, in case that h is a monotonic increasing function of ω(s,t), then the point patterns St are expected to exhibit an excess of clustering in areas with higher ω(s,t), thus positive association can be detected between the localized amount of clustering and estimated ω(s,t) (Watson, 2020). Firstly, we used the mean of the distances to the *K* nearest points (*D_K_*) to measure the clustering of locations, and calculated the rank correlation rt(K) between *D_K_* and the estimated ω(s,t) for survey year t. Here the estimated ω(s,t) was obtained from fitting the Bayesian spatial-temporal joint model. Next, the Monte Carlo method was used to sample realizations from the IPP under the null hypothesis (i.e., h≡0), following which, a set of rank correlations rt(K)M were calculated, approximating the distribution of the rank correlations ρt(K) under h≡0. In this way, the nonstandard sampling distribution of the test statistic can be approximated. Finally, we computed the desired empirical *p*-value by evaluating the proportion of the Monte Carlo-sampled rt(K)M which are more extreme than rt(K). We set a sample size of 1000 for each Monte Carlo sampling. We also considered *K* from 1 to 8 to measure the clustering of locations and resulted in eight *p*-values respective to different *K* for each survey year. If one of the *p*-values is smaller or equal to 0.05, we considered preferential sampling existing in the corresponding survey year. Since we modified our model to estimate the disease risk each year of the study period (please see the reply for the comment 2), this test was done for each survey year with number of locations more than or equal to 10. Results showed that significant preferential sampling might exist for survey locations in one third (6/18) of the survey years (Figure 2—source data 2). The corresponding impacts might include improper variogram estimator, biased parameter estimation and unreliable exposure surface estimates (Diggle et al., 2010, Pati et al., 2011; Gelfand et al., 2012). To avoid a more complex model, we didn’t take into account the preferential sampling issue for our final model, as the model validation showed a reasonable capacity of prediction accuracy. However, the disadvantage of this issue should be well aware. In response to the suggestion, we added the description of the test for preferential sampling in subsection “Model validation, sensitivity analysis and test of preferential sampling”, the test results in the Results, and the limitation and discussion in the Discussion.

2) The authors stratified predictions by 10-year periods; this is a very coarse time frame for predictions given that incidence of infection can vary from year to year. This limitation should be fully acknowledged by the authors if shorter time frames cannot be used.

We thank the reviewers for pointing out ways we could improve. We agree with your point of view. We improved the model by construction of the spatial-temporal random effects with temporal resolution each year instead of a 10-year period, which was able to estimate the disease risk yearly. We referred to the method proposed by Cameletti and colleagues (Cameletti et al., 2013; Krainski, 2019) to build a spatial-temporal model combined with covariates, which was defined as a SPDE model for the spatial domain and an AR1 model for the time dimension. To decrease the computational burden, under the SPDE framework, we built the GMRF on regular temporal knots, that is ω=(ωt=1978,ωt=1983,ωt=1988,ωt=1993,ωt=1998,ωt=2003,ωt=2008,ωt=2013,ωt=2018)′, while the latent fields corresponding to other years are approximated by projection of ω using the B-spline basis function of degree two, that is Bi,1(t)={1, ti≤t<ti+10, otherwise and Bi,m(t)=t−titi+m−1−tiBi,m−1(t)+ti+m−tti+m−ti+1Bi+1,m−1(t), where m is the degree of two (Cameletti, et al.,2012; Krainski,2019). We put more detailed description in subsection “Model fitting and variable selection”.

3) The reviewers criticized the imputation of sample size in order to convert prevalences to binomial data in papers where sample size was unavailable. While the authors included a sensitivity analysis of the impact of this imputation in Figure 3—source data 3 to help assess this point, this was not considered sufficient to address this issue. The reviewers suggest instead that if the data were originally available as prevalence estimates, these should be treated as such and modelled using a β likelihood or a normal likelihood (on the logit scale) and not converted artificially to binomial data.

We thank the reviewers’ comment and suggestion. For the survey data we collected, around 54.2% were reported with number of examined and number of positive, and the other 45.8% were only with the observed prevalence. Following the reviewers’ suggestion, in order to make full use of the available information (i.e., the reported the numbers of examined and the numbers of positive), we developed a bivariate model that jointly analyzes data reporting numbers of examined and positive, and data reporting only prevalence. Based on the probability theory, for data reported with numbers of examined and positive, we assumed that the number of examined Yit followed a binomial distribution Yit∼Bin(Nit,pit), where Nit denoted the number of examined; and for data only reported with the observed prevalence, we assumed that the observed prevalence obit followed a β distribution obit∼Be(pit,σβ2). Here pit was denoted the observed prevalence, number of examined, number of positive and the probability of infection, respectively. Furthermore, we modeled pit, the probability of infection, (from either types of distributions) in a logit form with same predictors and spatial-temporal random effects. Model validation showed that the performance of this model was satisfying, able to correctly estimate 79.61% of observations within a 95% coverage. We added the corresponding method in subsection “Model fitting and variable selection”, respectively.

4) The authors should describe how they dealt statistically when they encountered multiple estimates from the same area within each of the 10-year periods.

We thanked the reviewers for arising this point. In the revised manuscript, we modified the model for a yearly temporal resolution (please see the reply for the comment 2). In this case, we assumed the infection risk the same within 1-year period for the same areas. Different observations from the same year in the same areas can be treated as realizations of the randomized spatial-temporal process. Based on the fitted results, we estimated the infection risk each year of the study period at each pixel of the grid of 5×5km^2^ resolution. We added the corresponding descriptions in subsection “Model fitting and variable selection”, respectively.

5) Surveys often have complex designs, using weighting to calculate the prevalence over an entire area. How did the authors account for this weighting in their analysis?

We thank the reviewer for the comment on this important point. Indeed, for surveys across a large area, complex designs, such as randomly sampling from subgroups of the population under a well-designed scheme, are likely adopted, as it is impractical to draw simple random samples from the whole area. In such case, respondents may have unequal probabilities to be selected, thus weighting should be used to generalize results for the entire area. The observed disease data we collected were from surveys either at point-level (i.e., community or school) or aggregated over areas. For point-level data, as study areas were quite small, simple sampling design were mostly used in the corresponding surveys. And for areal-level data, particularly those aggregated across ADM1, complex designs were likely applied. However, most of the corresponding surveys were only reported raw prevalence or prevalence without clarifying whether weighting was applied. Thus, we did not have enough information to address the design effect for each single survey included. We put this limitation in the Discussion. On the other hand, as population density across the study region were different, we calculated the estimated country- and provincial level prevalence by averaging the estimated pixel-level prevalence weighted by population density, that is pp^A=∑i∈App^iwi/∑i∈Awi. Here pp^A, pp^i and wit are the estimated prevalence in area A, estimated prevalence at pixel i and population density at pixel i, respectively, where i belongs to area A. In this way, we took into account the diversity of population density across areas for regional summaries of the estimates. (subsection “Model fitting and variable selection”, and the Discussion).

6) The authors treated surveys aggregated over ADM2 or ADM3 areas as points, whereas those aggregated over ADM1 areas were treated as areal data. This is a very rough way to handle spatial misalignment. If the data were associated with areas, these should be left as areal data in the analysis and should not be treated as points as one would be enforcing non-existent geographical precision in the data in doing so. The authors should justify this choice, or discuss how it may impact the accuracy of results.

We thank the reviewers for pointing out ways we could improve. We agree with you, and treat all the survey data aggregated over ADM1, ADM2 and ADM3 as areal data. We have revised both the study protocol and results accordingly (Figure 1—figure supplement 1 and Figure 2).

7) The authors used the AUC statistic to validate their model. This is an inappropriate use of the AUC; ROC and AUC are normally used to check the discrimination ability of logistic regression models and not binomial regression models. The authors mention other metrics which are useful for evaluating binomial regression models such as MSE and MAE, but the values of these metrics are not discussed or presented in the manuscript. Please discard the AUC analysis, and instead include a table showing the values of these other metrics in the main manuscript, as well as the bias and 95% coverage rates of the fitted model.

We thank the reviewers’ suggestion. We modified the model validation part by using mean error (ME=1N∑(obit−ppit)), mean absolute error (MAE=1N∑|obit−ppit|), mean square error (MSE=1N∑(obit−ppit)2), as well as the coverage rate of observations within 95% BCI to evaluate the performance of the model. The ME, MAE, and MSE were 0.24%, 9.06%, and 2.38%, respectively, in the final model. And our model was able to correctly estimate 79.61% of locations within the 95% BCI, indicating the model had a reasonable capacity of prediction accuracy. We have revised both subsection “Model validation, sensitivity analysis and test of preferential sampling” and the Results of the manuscript accordingly.

8) The authors discuss differences in test sensitivity as a source of heterogeneity between surveys, which they ignored by assuming similar sensitivity across all surveys. It is unclear how much this may have affected results. Please give estimates of the magnitude of the difference of sensitivity of different diagnostic tests, as this could heavily influence differences in prevalence across surveys if these differences in sensitivity are very large. Is there a reason why the authors did not assess the diagnostic method as a covariate in their model?

We thank the reviewers’ comment. Previous studies have shown that the sensitivity and specificity of the same diagnostic method may differ across studies, while different diagnostic methods may result in different results in the same survey (Charoensuk et al., 2019; Laoprom et al., 2016; Sayasone et al., 2015). Due to the lack of enough information on the assessment of the quality and procedure of the diagnostic approach in each survey, we didn’t take into account this heterogeneity to the model in the original manuscript. Following the reviewers’ suggestion, by assuming the same diagnostic method has similar sensitivity and specificity across different surveys, we added the types of diagnostic methods, that is Kato-Katz, FECT and other methods (including methods other than the above two and methods not stated or missing) as covariates in the model, with Kato-Katz the baseline. The odds of infection differed significantly, with FECT resulted a lower odds than Kato-Katz, which was consistent with results found by Lovis and colleagues (Lovis et al., 2009). We have revised the subsection “Model fitting and variable selection”, results, Table 2 and discussion in the revised manuscript accordingly.

9) The authors used as an exclusion criteria surveys using the smear method to detect opisthorchiasis due to its lack of sensitivity. However, in nearly half of all reports, the diagnostic test used was not reported or missing. How do the authors then know that these records did not use the smear method to detect disease?

We thank the reviewers’ comment. As the direct smear has very low sensitivity, and only 5 relevant surveys used this method, we excluded them in the modeling analysis. However, there was a certain proportion of surveys (42%) with diagnostic techniques not stated or missing, and we were not able to know whether these surveys used direct smear as the diagnostic method, which was a limitation of the study. To partially taking into account the uncertainty, we considered the types of diagnostic methods as covariates in the modified model, grouping surveys with methods not stated or missing, or using methods other than Kato-Katz or FECT as the type “others”. The result shows that there was no significant difference between the odds of infection with other methods and that with Kato-Katz (Table 2, Results in revised manuscript).

10) The authors need to provide a list of citations of all their included studies as an appendix, consistent with GATHER item 5 and PRISMA item 18. GATHER also suggests providing a table with each data source used, reference information or contact name/institution, population represented, data collection method, year(s) of data collection, sex and age range, diagnostic criteria or measurement method, and sample size, as relevant.

We thank the reviewers’ suggestion. We provided a table, listing relevant information (reference, population represented, data collection method, year of survey, et al.,) for each data source in Figure 2—source data 1.

11) The interpretation of the estimated regression coefficients of the categorical variables was poorly done. In particular for model results Table 2: since the authors used a logit link function, the model results can be converted into odds ratios by exponentiating the model coefficients. Please convert all coefficients in this table into odds ratios. Model coefficients have very little inherent interpretability, while odds ratios can be interpreted by readers as measures of relative risk comparing the reference category and the category in question in relation to the outcome variable. The authors may also want to consider dropping the other non-coefficient model parameters from this table (spatial range, correlation coefficient, spatial variance) and report them in the text instead as their units are not consistent with the rest of the table. For the probability %, this would be reinterpreted as the probability that the odds ratio is >1 for risk factors increasing the prevalence of disease, and <1 for risk factors decreasing the prevalence of disease (distance to nearest open body of water and precipitation). Also, for the variables that were modeled as continuous (precipitation, HII), we need the unit size increase associated with each increase in prevalence (i.e. what increase in annual precipitation is associated with the 0.14 decrease in the logit?)

We thank the reviewer for these helpful suggestions. According to your suggestion, we revised the Table 2, by adding another column for the odds ratio (*OR*) and redefined the “Prob(%)” as the probability of *OR*>1. We also moved results of other non-coefficient model parameters to the text (Results). As we have modified the model according to reviewers’ suggestion, variable selection was re-run. And seven variables were selected for the final model, that is, survey type, diagnostic methods and land surface temperature (LST) in the daytime in categorical form, and human influence index, distance to the nearest open water bodies, elevation and travel time to the nearest big city in continuous form (Table 2 in revised manuscript). We added the interpretations of *OR*s for each covariate in the revised manuscript as following “The infection risk was 2.61 (95%BCI: 2.10-3.42) times in the community as much as that in school-aged children. Surveys using FECT as the diagnostic method showed a lower prevalence (*OR* 0.76, 95%BCI: 0.61-0.93) compared to that using Kato-Katz method, while no significant difference was found between Kato-Katz and the other diagnostic methods. Human influence index and elevation were negatively correlated with the infection risk. Each unit increase of the HII index was associated with 0.01 (95%BCI: 0.003-0.02) decrease in the logit of the prevalence. And increase of 1 meter in elevation was associated with 0.003 (95%BCI: 0.001-0.005) decrease in the logit of the prevalence.” (Results).

[Editors' note: further revisions were suggested prior to acceptance, as described below.]

The manuscript has been improved but there are some remaining issues that need to be addressed before acceptance, as outlined below:1) For Figure 5, negative values are conventionally interpreted as decreases and positive values as increases, so the numbers in this figure are likely to lead to confusion. Please change the calculations instead to (𝑝𝑝𝑠𝑡j − 𝑝𝑝𝑠𝑡i )/𝑝𝑝𝑠𝑡𝑖 , which should lead to an inversion of the sign without changing the numbers, and will increase the interpretability of the figure.

We thank the editors’ suggestion. We modified the calculations as (ppstj−ppsti)/ppsti. We have revised the method in the revised manuscript (subsection “Model fitting and variable selection”) and changed Figure 5 and Figure 5—source data 1 accordingly. And in the revised Figure 5, the red color represents increase of the risk, while the blue represents decrease of the risk.

2) In Table 2, the exponent of the intercept of the model cannot be interpreted as an odds ratio, as it represents the odds of the prevalence at the reference value of all categories. Please leave the cells for OR and prob(%) blank for this row, as these quantities are not relevant for the intercept.

We thank the editors for pointing out ways we could improve. Following the editors’ suggestion, we left the cells for *OR* and prob(%) blank for the row of the intercept (Table 2).